# Differential Effects of White Wine and Ethanol Consumption on Survival of Rats after a Myocardial Infarction

Danica Boban [1], Ana Marija Dželalija [2], Diana Gujinović [2], Benjamin Benzon [1], Nikola Ključević [1], Zvonimir Boban [3], Ivana Mudnić [2] and Ivica Grković [1,*]

1    Department of Anatomy, Histology and Embryology, University of Split School of Medicine, 21000 Split, Croatia
2    Department of Pharmacology, University of Split School of Medicine, 21000 Split, Croatia
3    Department of Medical Physics and Biophysics, University of Split School of Medicine, 21000 Split, Croatia
*    Correspondence: ivica.grkovic@mefst.hr; Tel.: +385-21-556525

**Abstract:** Studies of the cardioprotective effects of wine are mainly focused on red wines, due to their much higher content of bioactive compounds relative to white wines. Although some studies indicate a cardioprotective effect of white wine, there is no clear consensus on the existence of additional benefits of white wine over ethanol. The aim of this study was to determine and compare the effects of moderate consumption of white wine and ethanol on the survival of rats subjected to surgically induced myocardial infarction (MI). Male Sprague Dawley rats (n = 74) were randomized into three groups: water only, white wine or a 13% *v/v* ethanol/water solution. After a four-week drinking period, MI was induced by ligating the left anterior descending artery. The survival rate was highest in the wine group (72.2%), and lowest in the water only group (47.8%). There was no statistically significant difference in survival between the ethanol and water groups. An analysis linking drinking volumes to survival outcomes revealed that lower ethanol consumption was more prevalent in rats that survived, indicating an upper limit for the protective effects of ethanol. An opposite finding was noticed in the wine group, where no deaths occurred in rats with an average daily white wine consumption of approximately 10 mL or more. We conclude that moderate consumption of white wine has a positive effect on survival after a myocardial infarction, which cannot be attributed only to ethanol, but also to other white wine constituents.

**Keywords:** myocardial infarction; white wine; ethanol; rat; mortality; LAD ligation





## 1. Introduction

Despite the development of new strategies in the treatment of cardiovascular diseases (CVDs), they are still a leading cause of death around the world. As such, CVDs are the major causes of disease burden in Western and developing countries. Out of all deaths caused by CVDs, acute myocardial infarction (MI) accounts for 85% or 15.2 million deaths in total [1,2].

One of the risk factors for the development of ischemic heart disease is excessive alcohol consumption [3]. On the other hand, a J-shaped dependence of mortality on alcohol consumption was observed a century ago, showing that individuals drinking moderate amounts of alcohol lived longer than nondrinkers or heavy drinkers [4]. Since then, numerous epidemiological and, more recently, randomized controlled studies, have shown that moderate consumption of alcoholic beverages is associated with a lower risk of CVDs [5–13]. The protective effect of prior moderate alcohol consumption extends even to the post-recovery period, decreasing mortality after a MI [6,7,14,15]. Moreover, moderate consumption of alcoholic beverages may be beneficial even in patients who started drinking after recovering from a MI [6,7].

Having in mind the positive effect of moderate alcohol consumption, studies comparing the effects of various types of alcoholic beverages on human health highlighted the

beneficial effects of moderate consumption of wine [5,8,16]. Studies of the cardioprotective effects of wine, however, were mainly focused on red wines due to their much higher content of bioactive compounds when compared to white wines [17]. Although some studies have indicated cardioprotective effects of white wine [18,19], there is no clear consensus on the existence of additional cardioprotective effects of white wine over ethanol. This is in part due to epidemiological studies that generally do not differentiate between the two wine types or do not account for the ratios of red and white wine intake in subjects who drink both wine types.

The aim of this study was to find out whether prior moderate white wine consumption reduces mortality after a surgically induced MI in rats. In order to determine whether white wine constituents besides ethanol contribute to the result, in addition to water drinking controls and the white wine drinking group, we also included a group of animals drinking a water solution of ethanol. We also examined the relationship between the survival outcome and the amount of alcoholic beverage consumed by individual animals.

Controlled studies on humans focusing on the effect of alcoholic beverages on mortality after a MI are practically impossible to perform due to technical and ethical considerations. Consequently, such studies are mostly retrospective and based on patients' statements, making it hard to account for all confounding factors that might obscure the real effect. Therefore, an experiment using the proposed animal model under controlled interventional conditions would offer a clearer view of a cause–effect relationship.

## 2. Materials and Methods

### 2.1. Ethical Considerations

All procedures and experimental protocols were in accordance with the European Convention on Animal Protection and Guidelines on Research Animal Use and were approved by the Ethics Committee of the School of Medicine University of Split and by the Ethics Committee of the Ministry of Agriculture of the Republic of Croatia (No. 525-10/0255-16-7).

### 2.2. Animal Model and Experimental Design

For the study, we used 80 male Sprague Dawley (SD) rats, 10 to 12 weeks of age, weighing 300–350 g. After the exclusion of animals that refused to drink wine or died after anesthesia, seventy-four rats remained (Figure 1).

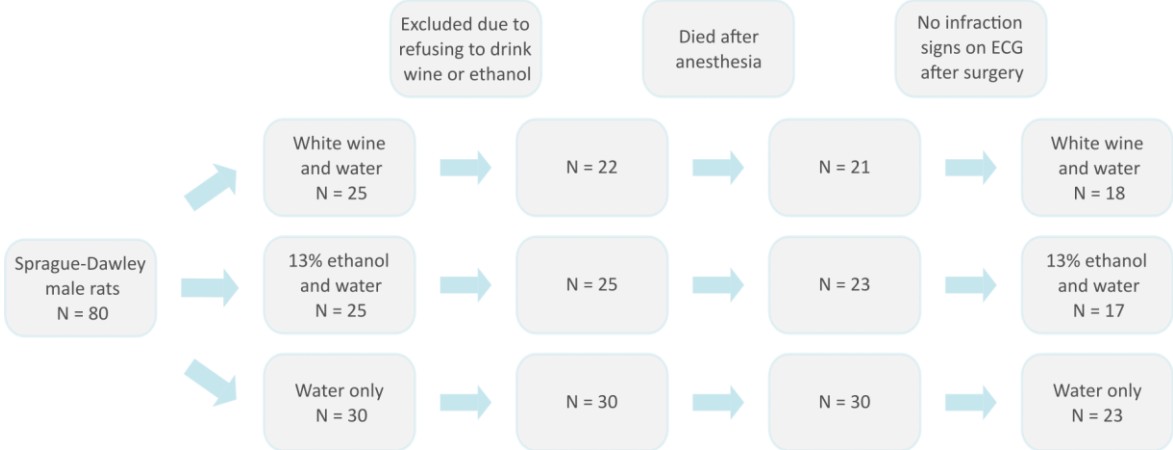

**Figure 1.** A flow diagram for rats included in the study.

All animals were raised in the Experimental Animal Facility of the University of Split. Ambient temperature was maintained at $23 \pm 2\,^\circ$C and on a 12–12 h light–dark cycle. Rats were kept individually in plastic cages covered with sawdust and fed with standard rat chow (4RF21 GLP, Mucedola srl, Settimo Milanese, Italy). Experimental animals were

randomized into three groups: water only, white wine or 13% *v/v* ethanol/water solution (Figure 2a). The white wine used in the study was Graševina 2015, Krauthaker winery, Croatia, containing 13% alcohol. A detailed analysis of the wine components has been reported in a previous article by our group [20]. Animals in the water group consumed water ad libitum. The wine and ethanol drinking groups of animals had access to the wine or ethanol solution ad libitum, 24 h/day for four weeks with the daily inclusion of tap water for 6 h. The fresh beverages were supplied daily in small pet containers with a leak-proof nozzle (Ferplast SmallPet Sippy Water Bottle, Castelgomberto, Italy). Fluid consumption was recorded every day. The MI was surgically induced by ligation of the left anterior descending artery (LAD), following four weeks of the drinking protocol. Survival rates were compared four days after surgery.

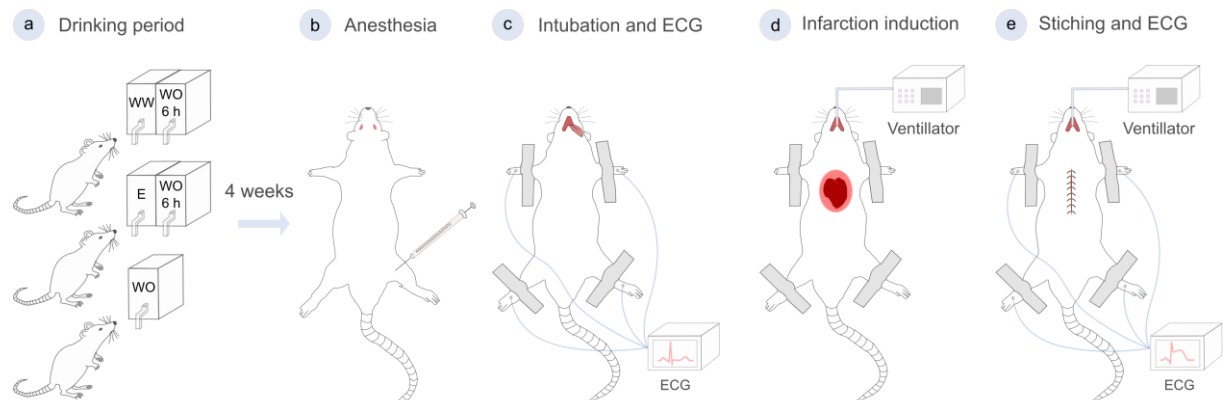

**Figure 2.** Workflow of the study. (**a**) Rats were divided into three treatment groups—a group drinking white wine (WW), a group drinking 13% ethanol (E) and a group drinking water only (WO). Aside from ad libitum access to alcoholic beverages, the wine and ethanol drinking groups had access to water for 6 h per day (WO 6 h). (**b**) Anesthetic was administered in the right hamstring muscle. (**c**) First ECG recording prior to surgery. (**d**) Surgically induced MI. (**e**) Second ECG recording in order to confirm the MI.

### 2.3. Surgical Procedure

Prior to surgery, anesthesia was performed using ketamine (Ketamine 10, 1.2 mL/kg; Intervent International, the Netherlands) and xylazine (Xylapan, 0.4 mL/kg; Vetoquintol, Bern, Switzerland) and was administered in the right hamstring muscle (Figure 2b). Rats were intubated with an 18-gauge arterial catheter. A 6-lead electrocardiogram (ECG) was then recorded as a reference for later comparison (Figure 2c). The surgery was performed through a midline incision of the abdominal wall using the "transdiaphragmatic approach" [21]. The xiphoid process was separated from the rest of the surrounding tissue and lifted up to visualize the abdominal surface of the diaphragm. A midline incision of the middle two quarters of the diaphragm was performed with the use of a surgical microscope (M520MC1; Leica, Switzerland). Animals were then connected to a ventilator. The pericardium was carefully removed and the LAD was ligated 2 mm under the left auricle using a 8.0-nonresorbable suture (Figure 2d). Successful ligation was identified by the presence of a pale area on the heart. Surgery was concluded by suturing the diaphragm and performing suction of the remaining air from the thoracic cavity (Figure 2e). Animals were then disconnected from the ventilator and continued to breathe spontaneously. After suturing the abdominal muscles and the overlying skin with a 4.0-nonresorbable suture, the ECG was recorded and compared with the first one. ECG findings, such as the presence of ST elevation, a new left bundle branch block, depression of the ST segment and inversion of the T-wave, were taken as a sign of myocardial infarction.

*2.4. Data Analysis*

The cutoff for statistical significance was $p < 0.05$. Student's *t*-test was used for comparison of means if the sample values were normally distributed. In the case of non-normal distributions, the Mann–Whitney U test was used. The Shapiro–Wilk test was used to assess the normality of the distribution. For comparison of proportions (survival rates), a hypothesis test based on a bootstrapped distribution was conducted. All data analysis was performed using the R programming language [22].

**3. Results**

*3.1. Survival Rates after Myocardial Infarction*

After excluding the experimental cases with unsuccessful MIs, based on the second ECG, we were left with 18 animals in the wine group, 17 in the ethanol group and 23 in the water drinking group (Table 1). The survival rate was highest in the wine group (72.2%), and lowest in the water only group (47.8%). Animals drinking ethanol were positioned between the other two groups with a survival rate of 64.7%. The survival rate in the wine drinking group was significantly greater than the one in the water drinking group ($p = 0.048$). There was no statistically significant difference between the water only and ethanol ($p = 0.143$), and ethanol and wine drinking ($p = 0.363$) groups (Figure 3).

**Table 1.** Number of rats per drinking group with corresponding survival rates 4 days after myocardial infarction.

| Treatment | Survived | Died | Total |
|---|---|---|---|
| White wine | 13 (72.2%) | 5 (27.8%) | 18 |
| Ethanol | 11 (64.7%) | 6 (35.3%) | 17 |
| Water | 11 (47.8%) | 12(52.2%) | 23 |

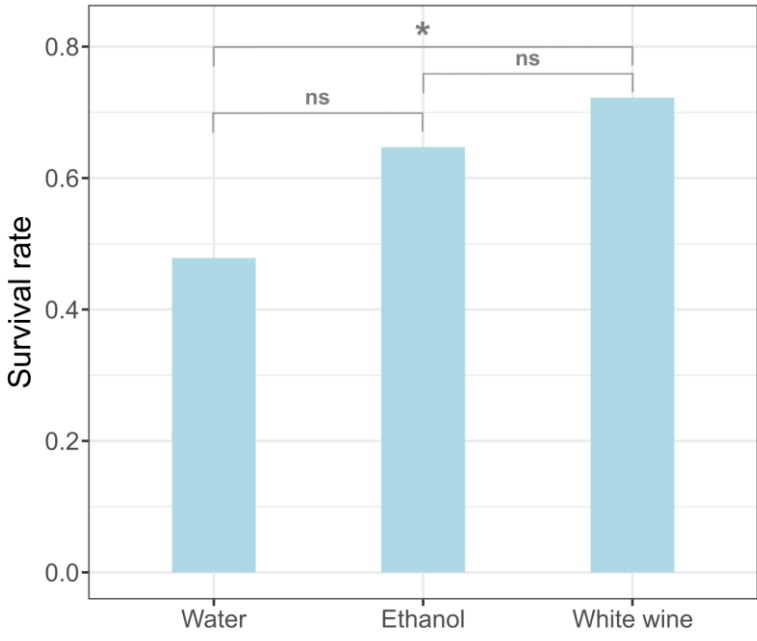

**Figure 3.** Comparison of survival rates between groups of rats drinking white wine, 13% ethanol or water only. NS = not significant. * denotes a *p*-value between 0.01 and 0.05.

*3.2. The Effect of Consumed Volume of Wine and Ethanol on Survival*

After comparing the survival rates between groups, we investigated the connection between the volume of consumed white wine and ethanol and the outcome for individual animals. The results are systematically presented in Table 2. We found that there was a

statistically significant difference between the consumed volume of ethanol in rats that died after MI and the rats who survived ($p = 0.003$).

**Table 2.** Intake of white wine, ethanol and water based on the outcome after MI. The last row contains *p*-values for tests of differences between the corresponding measurements within the same treatment group. The results are presented as mean $\pm$ SD. A *p*-value smaller than 0.05 was considered statistically significant. ns = not significant. ** denotes a *p*-value between 0.001 and 0.01.

| | White Wine | | | Ethanol | | |
|---|---|---|---|---|---|---|
| | Wine (mL) | Water (mL) | Mass (g) | Ethanol (mL) | Water (mL) | Mass (g) |
| Survived | $10.5 \pm 3.7$ | $20.1 \pm 1.2$ | $329 \pm 13$ | $11.0 \pm 2.0$ | $12.1 \pm 2.1$ | $330 \pm 37$ |
| Died | $8.21 \pm 0.93$ | $19.4 \pm 8.2$ | $351 \pm 11$ | $14.3 \pm 1.6$ | $11.9 \pm 2.3$ | $345 \pm 1$ |
| Significance | ns | ns | ns | ** | ns | ns |

A more detailed inspection showed that higher ethanol consumption was associated with a negative outcome (Table 2, Figure 4). In contrast to the ethanol group, there were no deaths recorded in rats that drank an average daily amount of approximately 10 mL or more of white wine (7 out of 13 rats who survived) (Figure 4). The average daily water consumption and average mass before surgery were not related to the difference in outcomes (Table 2).

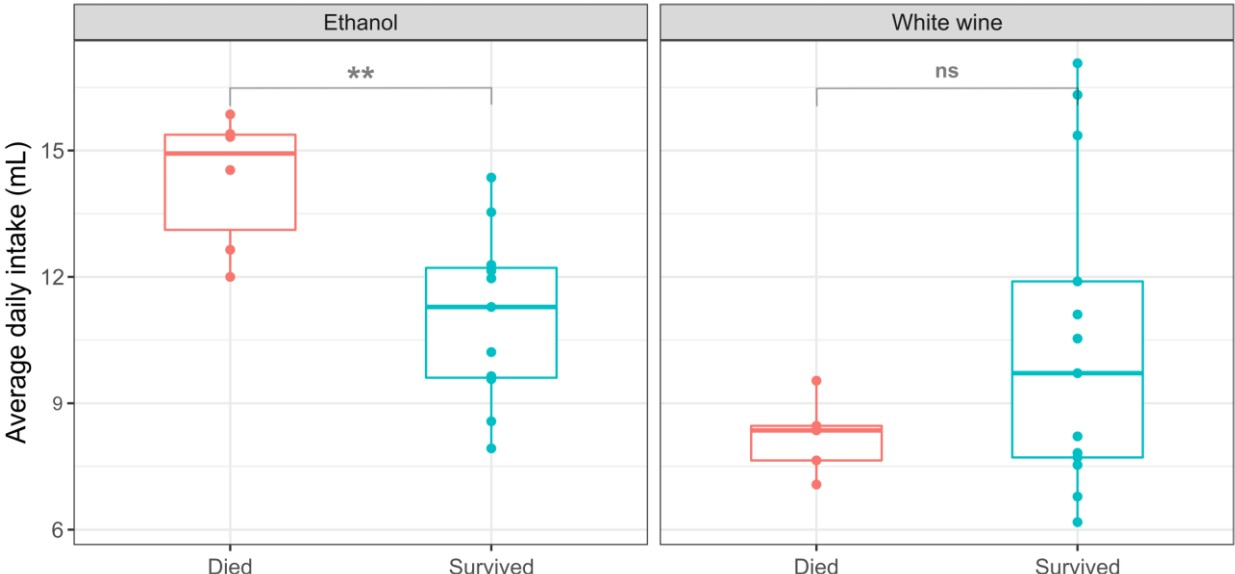

**Figure 4.** Comparison of daily intake of ethanol and white wine depending on the outcome (died or survived) four days after induction of MI. ns = not significant. ** denotes a *p*-value between 0.001 and 0.01.

## 4. Discussion

The key finding of this study is that rats drinking white wine for 4 weeks prior to a surgically induced MI had significantly better survival than control animals drinking only water. In contrast to the wine group, consumption of a corresponding amount of ethanol did not result in significantly better survival relative to the water group. In addition, a more detailed analysis linking drinking volumes to survival outcomes revealed that lower ethanol consumption was more prevalent in rats that survived. These results indicate an upper limit for the protective effect of ethanol. Contrary to the ethanol drinking group, there seems to be no such limit for wine, as no deaths occurred in rats with an average daily white wine consumption of approximately 10 mL or more. This implies that the

cardioprotective effect of white wine cannot be attributed solely to ethanol, but to other wine constituents as well. The fact that we were able to separate outcomes based on different levels of ethanol intake indicates the suitability of the experimental model used in terms of the intensity of the pathophysiological challenge and its sensitivity in detecting the effects of different drinking protocols.

It should be pointed out that the intake of both ethanol and white wine by the animals in our study falls into the low to moderate drinking regime [20,23]. Nonetheless, an upper limit for protective effects was observed in the ethanol group, indicating a hormetic physiological behavior of ethanol in which the effects of small quantities are opposite to that of large quantities. In other words, low-intensity stress in a living system provoked by low doses of ethanol triggers a series of countering mechanisms to ensure subsequent protection and homeostasis [24,25]. Indeed, depending on the amount consumed, ethanol may increase or decrease the risk of CVDs, reflecting a J-shaped relationship between CVDs and alcohol intake [26]. It has been well established that the beneficial cardiovascular effects of moderate alcohol consumption can partly be attributed to alcohol-associated increases in both HDL cholesterol and fibrinolytic activity, as well as decreased platelet aggregation [27]. In the case of wine, cardioprotective effects may be attributed to both alcoholic and non-alcoholic bioactive compounds, primarily wine polyphenols and, possibly, other bioactive phytochemicals such as melatonin and phytosterols [28,29]. Because of the positive effects of low amounts of alcohol on HDL-cholesterol and hemostatic factors, ethanol and polyphenols may exert additive and/or synergistic protective effects on the cardiovascular system [30].

Due to the differences in wine-making techniques, the total phenolic content in standard white wines is much lower than in red wines. Nevertheless, white wines still contain a variety of phenolics, particularly hydroxycinnamic acids and derivatives, together with a range of other bioactive products from the microbial metabolism, such as tyrosol, hydroxytyrosol, or melatonin, which may be responsible for the superior beneficial effects of white wine over ethanol [31]. Studies with some of these white wine components in animal models have shown beneficial effects in the management of oxidative stress and inflammatory reaction. Samuel et al. have shown that n-tyrosol-treated rats with an induced MI display a reduced infarction size and cardiomyocyte apoptosis along with an improvement in the myocardial functional parameters [32]. Moreover, it has also been demonstrated that infusion of n-tyrosol prior to inducing MI in rats reduces the incidence and severity of ventricular arrhythmias during acute myocardial ischemia and reperfusion [33]. Caffeic acid, a typical member of hydroxycinnamic acids, exhibited blood pressure-lowering properties and reduced activities of key enzymes linked to the pathogenesis of hypertension in cyclosporine-induced hypertensive rats [34].

Although the mechanisms underlying the biological effects of polyphenols are still a matter of debate, there is considerable evidence that, on a cellular level, the major effect of phytochemicals found in wine includes activation of a signal transduction pathway, resulting in an increase in the antioxidant enzymes and their substrates. In that way, phytochemicals from wine may increase the cellular resilience to oxidative stress. This phenomenon involves an apparently paradoxical effect in which the phytochemicals are metabolically converted to electrophiles. These are capable of triggering a hormetic-like response that does not require the initiation of stress as is the case with low-dose toxicants [35–37]. By acting in the described way, polyphenols from wine may counteract the detrimental effects of ethanol and its toxic metabolites.

It is important to understand that the cardiovascular benefits of moderate wine consumption may also occur indirectly due to the effects on other organs or organ systems such as protection against postprandial oxidative/inflammatory stress [38] or prebiotic effects on favorable gut microbiota composition [39].

## 5. Conclusions

This is one of few experimental studies examining the biological effects of white wine. It clearly demonstrates its protective effect under the severe pathophysiological challenge of surgically induced MI. The survival rate of rats after an induced MI was significantly higher in rats that drank wine (72.2%) compared to the water drinking group (47.8%). Moreover, white wine appears to be safer and more effective in that regard than comparable amounts of ethanol. Considering the amount of beverage consumed, an upper limit for the protective effect of ethanol was observed, but not for wine. In order to deepen the insight into the processes associated with the observed protective effects of white wine consumption, we plan to perform further studies on the parameters of healing after MI. The increasing popularity of white wine in different dietary cultures, and perspectives of increasing demand on a global scale [40], justify the need for more research on the biological effects of white wine. We hope that this simple, yet straightforward, study will serve as motivation for further research that will provide more insight into this complex matter.

**Author Contributions:** Conceptualization: I.G. and I.M.; methodology: I.G., D.B., B.B. and N.K.; investigation: I.G., D.B., B.B., N.K., A.M.D., D.G. and Z.B.; data curation: D.B.; formal analysis: Z.B. and D.B.; writing—original draft preparation: D.B.; writing—review and editing: I.G., D.B. and Z.B.; visualization: D.B. and Z.B.; supervision: I.G. All authors have read and agreed to the published version of the manuscript.

**Funding:** This study was funded by the Croatian Science Foundation (Project no. 8652), whose support is gratefully acknowledged.

**Institutional Review Board Statement:** All procedures and experimental protocols were in accordance with European Convention on Animal Protection and Guidelines on Research Animal Use and were approved by the Ethics Committee of the School of Medicine University of Split and by the Ethics Committee of the Ministry of Agriculture of the Republic of Croatia (No. 525-10/0255-16-7).

**Informed Consent Statement:** Not applicable.

**Data Availability Statement:** The data presented in this study are available upon reasonable request from the corresponding author.

**Conflicts of Interest:** The authors declare no conflict of interest.

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
