# Peer review of "Differential Effects of White Wine and Ethanol Consumption on Survival of Rats after a Myocardial Infarction"

_applsci, doi:10.3390/app13031450_

Round 1

Reviewer 1 Report

A major concern I have is the applicability to humans. Is there data that the metabolism and impact of alcohol in rats comparable in man. Also, one month of alcohol may ne be equivalent to many years of alcohol consumption. that needs to be addressed.

Author Response

Dear reviewer 1:

We have carefully considered your suggestions and remarks. Please find attached our point-by-point response to your comments and accordingly revised manuscript.
We hope that you will find our answers to your comments satisfactory and that the paper in this form will be acceptable for publication in the Applied Sciences.

Reviewer 2 Report

Dear Authors,

in the attached manuscript you will find my proposed revisions. The paper is well written, but some essential points should be revised, after which this paper can be accepted in my opinion.

Good work

Author Response

Dear reviewer 2:

We have carefully considered all your suggestions and remarks. Please find attached our point-by-point response to your comments and accordingly revised manuscript.
We hope that you will find our answers to your comments satisfactory and that the paper in this form will be acceptable for publication in the Applied Sciences.
